Development and validity of the expectations of physiotherapists questionnaire on practice management software

Arza-Moncunill Eduardo 1
Medina-Mirapeix Francesc mirapeix@um.es 2
Martín-San Agustín Rodrigo 1
1 Department of Physiotherapy, University of Valencia , Valencia , Spain
2 Department of Physiotherapy, University of Murcia , Murcia , Spain
La Touche Roy
Electronic publication date: 2023 Oct 17
Publication date: 2023
Volume: 11
Electronic Location ID: e16246
Received 2023 Mar 23; Accepted 2023 Sep 14
Copyright: ©2023 Arza-Moncunill et al.
Copyright year: 2023
Copyright holder: Arza-Moncunill et al.
License: This is an open access article distributed under the terms of the Creative Commons Attribution License, which permits unrestricted use, distribution, reproduction and adaptation in any medium and for any purpose provided that it is properly attributed. For attribution, the original author(s), title, publication source (PeerJ) and either DOI or URL of the article must be cited.
License URL: https://creativecommons.org/licenses/by/4.0/

Keywords: Validity, Questionnaire, Practice management software, Physiotherapists

Funding: The authors received no funding for this work.

==============================
Background

Despite the growing trend in the use of digital technologies in physiotherapy, the overall adoption of both, practice management software (PMS) and electronic health records in physiotherapy clinics has been low and slow over time. In order to learn what factors determine the adoption of these technologies by physiotherapists, there is a need to examine the expectations of physiotherapists (EPs) on specific software attributes. The main aims of this study were to develop a questionnaire to measure and describe the EPs towards PMS. The knowledge of these EPs will be useful to guide PMS design in order to improve physiotherapists‘ satisfaction.

Methods

Instrument development study with validity and reliability testing. The development of this questionnaire was conducted in three phases: identification of attributes to be explored, development of the items, pilot study, and psychometric testing. The questionnaire was distributed to chartered physiotherapists. A total of 272 participants completed the questionnaire.

Results

A series of analysis were conducted to assess item reduction, factor structure of the questionnaire and metric properties of multi-item scales. From the initial 43 attributes, the final version of the questionnaire consisted of 26 items on EPs, grouped in nine scales and two areas (clinical care and administrative activities). As a result, all scores had strong item-scale correlations, excellent item scaling success, and good internal consistency (Cronbach alpha coefficients of >.7). Our study also showed that current EPs were high towards most of the items, only two scales concentrated most of the attributes with the least expectations (monitoring quality of care and digital health interventions). Our study included physiotherapists with and without experience with PMS, and it showed that both groups had a similar pattern of expectations. Our study provides a valuable questionnaire of EP on PMS attributes for clinical care and administrative activities and shows a detailed development process.

Introduction

Institutions like the United Nations, the World Health Organization or the European Union have recognized the value of digital technologies to improve healthcare delivery (Gagnon et al., 2009; Peterson, Hamilton & Hasvold, 2016; World Health Organization. Regional Office for Europe, 2018; Bonacina et al., 2021) due to advantages such as improving the accessibility and exchange of information, healthcare quality and safety, or the efficiency and productivity among others (McGinn et al., 2011; Nguyen, Bellucci & Nguyen, 2014; Peterson, Hamilton & Hasvold, 2016). In particular, there has been a growing trend in physiotherapy in the use of practice management software (PMS) which focus on practice management aspects of the business (e.g., appointment booking, billing, and accounting; Physiotherapy Association of British Columbia, Canadian Physiotherapy Association), electronic health records functionalities (e.g., medical history, images, test results, and treatment plans), or telerehabilitation (Vreeman et al., 2006), considered as an alternative to usual face-to-face treatments with benefits in multiple pathologies (Rausch et al., 2021). In spite of that positive trend towards these technologies, their overall adoption in the physiotherapy clinics has been low and slow over time (Vreeman et al., 2006; Postolache et al., 2015; Messer-Misak & Egger, 2016; Postolache, Oliveira & Postolache, 2017; Yung, 2017).

During the past decade, there has been an interest in learning what factors determine the adoption of these technologies by physiotherapists (Vreeman et al., 2006). Such interest has further increased since the COVID-19 pandemics (Rausch et al., 2021; Reynolds, Awan & Gallagher, 2021). Most studies about this topic used cross-sectional surveys (Messer-Misak & Egger, 2016; Postolache, Oliveira & Postolache, 2017; Rausch et al., 2021; Reynolds, Awan & Gallagher, 2021), and the questionnaires often addressed experiences in the use of digital technologies (Messer-Misak & Egger, 2016; Postolache, Oliveira & Postolache, 2017), attitudes towards technologies (Rausch et al., 2021), requirements for the PMS (Australian Physiotherapy Association (APA), 2018) and/or physiotherapists (e.g., training, formation) (Rausch et al., 2021), beliefs on benefits and barriers of their adoption (Tanriverdi & Iacono, 1999; Rausch et al., 2021; Reynolds, Awan & Gallagher, 2021), patients’ satisfaction (Laver et al., 2012; Reynolds, Awan & Gallagher, 2021), and physiotherapists’ job satisfaction (Reynolds, Awan & Gallagher, 2021).

As a consequence of all these studies, multiple recommendations have been proposed to reduce barriers and to increase positive attitudes and satisfaction with digital technologies (Vreeman et al., 2006; McGinn et al., 2011; Rausch et al., 2021; Reynolds, Awan & Gallagher, 2021). These recommendations mainly focused on the design of an adequate software, which should meet or exceed physiotherapists’ expectations to avoid dissatisfaction with its use (Vreeman et al., 2006). In spite of the apparent relevance of expectations, which is a belief that an object possesses or should possess a particular attribute (Cardello, 2007), expectations of physiotherapists (EPs) have received scarce attention. While other theories were closely examined by specific questionnaires, minor approaches have been made to examine EPs on specific PMS attributes (e.g., software use for the therapeutic process; Messer-Misak & Egger, 2016). Moreover, there is a lack of specific questionnaires. Considering this, a questionnaire created to examine EPs on specific PMS attributes could be useful to guide PMS design in order to improve physiotherapists‘ satisfaction, and thus, facilitate its implementation in clinics.

Thus, the main aims of this study were to develop a questionnaire to measure EPs and to describe the EPs towards PMS. The questionnaire will be named “Expectations of Physiotherapists Questionnaire on Practice Management Software” (EPQPMS).

Materials & Methods

The development of this questionnaire was conducted in three phases: (I) identification of the attributes to be explored on the basis of a literature review and a qualitative study; (II) items for each one of the identified attributes were developed and tested for clarity and relevance through cognitive pretesting with potential respondents and focus group participants; (III) the questionnaire psychometric properties were examined in a survey among physiotherapists.

Phase 1: identification of attributes to be explored

Several steps were taken to get a comprehensive overview of this subject and to identify the attributes to be explored. Firstly, a literature review of the relevant publications was carried out. The most relevant national and international management programs, identified in consultations with national and international physiotherapy associations, were studied to learn the main features and conditions. Secondly, six focus groups were conducted including physiotherapists with knowledge and experience on this topic to identify the needs on PMS. Videotape, audio, and field notes were used for data collection. Finally, a topic guide was developed based on the theoretical framework. The results of this phase are reported elsewhere (in revision). In summary, twelve subthemes of attributes grouped in two areas were identified: clinical care and administrative activities. In the clinical care area, the subthemes focused on templates, digitalized tools, and classification codes for data entry; digitalized tools and templates for the issuance of individualized patient reports; digitalized measures to monitor quality of care; automatized reminders for patients and professionals within the scheduling agenda; digital health interventions (DHI); and patient portal. In the administrative activities area, the subthemes focused on automatized issuance of routine documents; communication tools for marketing strategies; flexible billing and automatized accounting of payments; automatized accounting of supplies to control stock; interoperability; data security to face failures, inadequate use and threats.

Phase 2: development of the items

The objective of this phase was to construct questions based on the attributes identified. The precise wording of the items was based on the physiotherapists’ comments in the focus groups and modified through a process of discussion and consensus among the research team. The questions were presented in an array format. Each area included an overall question that was phrased, “The desirable software for your center should allow (the following attributes)”.

A 7-point Likert scale was used for the responses, ranging from “very much disagree” to “very much agree”. An initial pool of 43 items were phrased, 22 in the subthemes of the clinical care area and 21 in those of the administrative activities area.

The questionnaire was pretested for clarity and adequacy of content with 11 physiotherapists in two focus groups. Four questions were rephrased because they were leading to misunderstandings. Thus, in two items, five of the physiotherapists expressed doubts about the examples used to explain the item. In the other two items, four and three physiotherapists did not fully understand any of the functions that the item wanted to express. The reformulation of these items, which were also items that were maintained in the final questionnaire, are included in Appendix S1. The instrument finally distributed for evaluation comprised 43 items of EPs on management software. Additionally, it included socio-demographic data.

Phase 3: pilot study and psychometric testing

In phase 3, psychometric testing of the questionnaire was carried out, surveying physiotherapists in the Valencian Community during 6 months by completing the online questionnaire on Limesurvey. The link to the questionnaire was distributed by the ICOFCV (Illustrious Official College of Physiotherapists of the Valencian Community) to all chartered physiotherapists via email and posted on their corporate social networks with the pertinent explanations. Eligible participants were all graduated physiotherapists who had enough digital skills to fill in the online questionnaire in order to learn their expectations on PMS. Reminders were also sent 1 and 3 months later. The answers were totally anonymous and a cookie was set to avoid repeated participations.

Data analysis

To describe the participants’ socio-demographic characteristics and EP, descriptive statistics were used. For each of the two areas (clinical care and administrative activities), a series of analysis were conducted to assess item reduction, factor structure of the questionnaire and metric properties of multi-item scales. All analyses were performed with the SPSS 28.0 (SPSS Inc., Chicago, IL, USA).

For a first item selection, variance and non-response rates for items were evaluated.

As DeVellis suggests, items that were poorly understood [i.e., with high rates of non-responses (≥10%)] or items with poor variability were eliminated (DeVellis, 2016), for the latter a standard deviations of less than 0.60 was selected, as it was done elsewhere (Medina-Mirapeix et al., 2015). As previously described in Medina-Mirapeix et al. (2015), exploratory factor analyses were performed to identify latent factors that could be responsible for the covariation of the data. Principal components analysis and varimax rotation were used for the initial extraction of factors. Items with loadings of 0.50 or higher were retained, but items with factor loading of less than 0.40 on one factor or higher than 0.40 on more than one factor were removed (Ware & Gandek, 1998). A parallel factor analysis was also performed using the maximum likelihood and principal axes methods to check if there was consistency in the identified factors. In case of non-consistency, such factors were eliminated.

In a multi-trait scaling analysis, a correlation matrix of all items of each area and scale was used to test the extent to which items converge with and diverge from scales. Scales were scored with a method of averages, summing up the scores of each item in the corresponding scale and dividing it by the number of items in that scale. Correlations between each item and its hypothesized scale were calculated and corrected for overlap by not including them in the scale. Correlations of 0.40 or higher were considered to be satisfactory; items with correlations of less than 0.40 were removed for further analyses (Briggs & Cheek, 1986).

Following what was described by Ware & Gandek (1998) and as previously carried out by Medina-Mirapeix et al. (2015), scaling success rates were computed for each scale as the percentage of items within the scale that correlated more highly or significantly more highly with the hypothesized scale than with the other scales. An item correlated significantly more highly with its own scale if the correlation between the item and its hypothesized scale was more than two standard errors higher than its correlation with other scales (Ware & Gandek, 1998). In addition, to test the internal consistency reliability, the Cronbach alpha coefficient was calculated for each, >0.7 and >0.9 were considered acceptable and excellent, respectively (De Vet et al., 2011).

Finally, boxplots were created to describe the distribution of expectation scores for each attribute by area. Also, we calculated the percentages of physiotherapists who expected the presence of all items on a scale and those who did not expect any. Student’s t test was used to compare the percentages between physiotherapists with or without experience with PMS.

Results

A total of 272 physiotherapists completed the questionnaire. The participation included a similar proportion of men and women, 13 years of experience in average and a clear predominance of the private sector. Remarkably, only 54% of participants had experience with a PMS solution in work. The participants’ characteristics are shown in Table 1.

Table 1 Characteristics of participants (n = 273).

Sex	Men	48.5%	
Women	51.5%	
Age	<35	39%	
35–50	53.3%	
>50	7.7%	
Workplace	Physiotherapy clinic	74.7%	
Hospital 2	3.9%	
Primary health centres	5.8%	
Day care centre/Residence	2.6%	
Sports club/centre	5.8%	
Others	7.2%	
Job Position	Owner/Manager	40%	
Employee in a private centre	27.7%	
Self-employed	25.2%	
Employee in a public centre	7.1%	
Years of experience	Average	13 years	
Years working in that centre	Average	8.4 years	
N° of physiotherapists working at the centre	1	29%	
2 or 3	40%	
≥4	31%	
Clerk staff	Yes	49%	
No	51%	
Experience with PMS	Yes	54%	
No	46%	

As previously indicated, the initial questionnaire consisted of two areas, divided into six subthemes each, with 22 items in the clinical care area and 21 in those of the administrative activities area.

In the clinical care area, 10 items were eliminated in different steps: (1) in a first principal components analysis, several items showed factor loading higher than 0.40 on more than one factor: four items initially grouped in the subtheme “data entry and issuance of reports” (factor loading range from 0.417 to 0.610), and one item from the subtheme “DHI” (factor loading in two components of 0.421/0.570); (2) in a second main components analysis, one item from the subtheme “monitoring quality of care” and one item from “patient portal” showed a loading factor in two components of 0.428/0.674 and 0.401/0.557, respectively, (3) in a third main components analysis, one item from the subtheme “workflow coordination” showed a factor loading in two components of 0.444/0.670, and (4) the subtheme “workflow coordination”, with two items, was eliminated due to inconsistency among the three factor analysis methods: in principal components method a factor loading of 0.798/0.805, in principal axes method a factor loading of 0.396/0.448, and in maximum likelihood method a factor loading of 0.342/0.372. Thus, for final factor analysis we had four scales and 12 items left in this area. The final result of the factor analysis of the complementary methods (i.e., principal axes and maximum likelihood methods) can be seen in Appendix S2.

In the administrative activities area, seven items were eliminated in different steps: (1) in a first principal components analysis, two items initially grouped in the subtheme “issuance of routine documents” and two items in the subtheme “billing and accounting” showed factor loading higher than 0.40 on more than 1 factor (factor loading range from 0.434 to 0.627 and 0.416 to 0.621, respectively), (2) in a second principal components analysis, one item in the subtheme “data security” showed a loading factor in two components of 0.479/0.611, and (3) the two items in the subtheme “interoperability” initially grouped in the principal components analysis with items from “Data security”, were eliminated for correlating less than two standard errors on that scale (r = 0.610 and r = 0.541) versus “issuance of routine documents” (r = 0.556 and r = 0.501). Thus, for final factor analysis we had five scales and 14 items left in this area.

In resume, from the initial 12 subthemes, only two subthemes (workflow coordination and interoperability) were excluded from the questionnaire, while two subthemes emerged into a single subtheme (data entry and issuance of reports).

Structure of the questionnaire and metric properties

Tables 2 and 3 show how the items included in the final factor analyses of each area loaded significantly onto their scales and how they were named.

Table 2 Factor analysis of 12 items of the clinical care area.

		Value for factor	
Scales	Items	1	2	3	4	
Data entry and issuance of reports	Editable body charts	0.816	0.171	0.128	0.098	
Digitalized patient-reported outcome measures	0.815	0.238	0.091	0.094	
Templates for patients’ clinical reports	0.723	0.209	0.191	0.057	
Editable templates for assessment	0.710	0.275	0.162	0.236	
Templates for exercise programmes and recommendations	0.670	0.154	0.101	0.169	
Monitoring quality of care	Quality measures	0.268	0.855	0.188	0.228	
Healthcare activity reports	0.265	0.829	0.080	0.087	
Patient safety reports	0.341	0.793	0.198	0.209	
Digital health interventions	Videoconference	0.190	0.105	0.868	0.184	
Chat	0.211	0.239	0.862	0.096	
Patient portal	Online appointment booking	0.143	0.167	0.109	0.890	
Consult scheduled visits	0.230	0.203	0.169	0.826	
Notes.

Bold type indicates primary loading for each item.

Table 3 Factor analysis of 14 items of the administrative activities area.

		Value for factor	
Scales	Items	1	2	3	4	5	
Issuance of routine documents	Templates for common documents	0.186	0.859	0.217	0.145	0.079	
Automate the issuance of routine documents	0.224	0.829	0.211	0.116	0.076	
Easily fill in and sign documents for patients and professionals	0.059	0.594	0.332	0.105	0.246	
Data security	Saving and backup copies	0.229	0.329	0.779	0.077	0.118	
Security measures against computer threats	0.302	0.257	0.757	0.099	0.069	
Configuration of users and access permissions	0.019	0.145	0.756	0.214	0.203	
Billing and accounting	Fees configuration	0.843	0.197	0.194	0.172	0.185	
Flexibility in the application of fees	0.807	0.082	0.121	0.239	0.075	
Allow different payment methods	0.723	0.279	0.191	0.177	0.314	
Marketing strategies	Repository of standard messages	0.252	0.138	0.196	0.811	0.06	
Allows mass mailings of communications	0.094	−0.049	0.205	0.752	0.287	
Links to external communication applications	0.235	0.349	−0.016	0.736	−0.021	
Control stock supplies	Stock reports	0.202	0.145	0.143	0.117	0.907	
Notifications to replenish consumables	0.189	0.131	0.170	0.127	0.901	
Notes.

Bold type indicates primary loading for each item.

Generally, the multi-trait scaling analysis supported the scaling of items into all the hypothesized scales (Table 4). On the one hand, in the clinical care area, the item-scale correlations ranged from 0.764 to 0.937 with a percentage of scaling success of 100% for all the scales (Table). On the other hand, in the administrative activities area, the item-scale correlations ranged from 0.768 to 0.967 with a scaling success percentage of 100% for all the scales.

Table 4 Summary of Results for Multitrait Scaling Analysis.

Scale	Item-scale correlation range (median)	% Scaling success	Cronbach alpha	
Clinical care area				
Data entry and issuance of reports	0.781–0.842 (0.816)	100	0.873	
Monitoring quality of care	0.764–0.788 (0.776)	100	0.702	
DHI	0.910–0.937 (0.923)	100	0.822	
Patient portal	0.908–0.931 (0.919)	100	0.814	
Administrative activities area				
Issuance of routine documents	0.768–0.899 (0.848)	100	0.808	
Data security	0.852–0.862 (0.845)	100	0.780	
Billing and accounting	0.785–0.854 (0.825)	100	0.742	
Marketing strategies	0.860–0.905 (0.878)	100	0.842	
Control stock supplies	0.967–0.866 (0.915)	100	0.930	

The final version of the EPQPMS consisted of 26 items structured in two areas, 12 items in four scales in the clinical care area and 14 items grouped in five scales in the administrative activities area (see Appendix S3). Their reliability was acceptable to excellent, with Cronbach alpha coefficients ranging from 0.702 to 0.930 (Table 4). Considering subgroups between physiotherapists, with and without experience with PMS, Cronbach’s alpha coefficients were also higher than 0.700.

Description of the expectations for each attribute (item)

Figs. 1 and 2 show the distribution of the agreement in the scores reported in each one of the 26 attributes explored. For example, in the two items of the scale “DHI” a 25% of participants reported scores <4 (i.e., they did not have any degree of expectations in these items), and the remaining 75% reported some level of expectation (25% with high agreement scores ≥ 6). In most of the items at least 50% of participants reported some level of expectation (i.e., scores ≥ 4 or agree) and at least a 25% showed the highest score of 7 (maximum level of agreement). Regarding the floor effect, on the one hand, in nine of the 12 clinical items, less than 3% of the subjects scored 1 (in the other three items, around 6% of the subjects). On the other hand, in 12 of the 14 administrative items, less than 2% of the subjects scored 1 (in the other two items, around 5% of the subjects).

Figure 1 Scores of the 14 items in the clinical care area.

A score of 1 means “very much disagree” and 7 “very much agree” with each attribute being present in a PMS.

Figure 2 Scores of the 14 items in the administrative activities area.

A score of 1 means “very much disagree” and 7 “very much agree” with each attribute being present in a PMS.

Table 5 shows the percentages of physiotherapists who expected the presence of all items on a scale in the PMS and those who did not expect any, both in total and grouped according to whether or not they had experience with PMS. While four of the five administrative activities area scales showed a low percentage (<6%) of physiotherapists not expecting to find any of the items, this only happened in two of the four clinical care area scales, where “Monitoring quality of care” and “DHI” scales showed percentages superior to 18% of physiotherapists who did not expect to find any of their items in the PMS. Similarly, percentages greater than 75% of physiotherapists who expected to find all the items of four out of five scales in the administration activities area were observed. In addition, having or not having PMS experience did not show significant differences in any of the percentages.

Table 5 Percentage of physiotherapists who reported expectations in all items of each scale and who did not report any.

	Expected the presence of none of the items	Expected the presence of all items	
Scale	All PTs (n = 272)	All PTs (n = 272)	PTs with PMS experience (n = 147)	PTs without experience (n = 122)	Difference between groups p value	
Clinical care area						
Data entry and issuance of reports	3.3%	74.3%	75.5%	72.6%	0.636	
Monitoring quality of care	18.0%	61.5%	63.3%	61.6%	0.113	
DHI	35.5%	42.1%	40.1%	46.4%	0.331	
Patient portal	15.8%	65.5%	60.5%	72.0%	0.015	
Administrative activities area						
Issuance of routine documents	1.8%	87.1%	91.8%	84.8%	0.163	
Data security	1.8%	84.9%	87.1%	82.1%	0.266	
Billing and accounting	5.9%	79.8%	81.0%	79.0%	0.944	
Marketing strategies	4.8%	61.5%	62.6%	60.8%	0.385	
Control stock supplies	16.5%	76.8%	77.6%	76.6%	0.778	

Discussion

To the best of our knowledge, this is the first study to develop a questionnaire aimed to examine EP towards a list of attributes for PMS. The present study provided preliminary evidence on the reliability and validity of the fixed-length EPQPMS. Nine scales pertaining to two different areas, clinical care and administrative activities could be computed. Additionally, our study showed that current EPs were high towards most of the items, and that many physiotherapists showed expectations on all items of each scale.

Our first phase to identify the attributes to be explored by the EPQPMS was based on a qualitative research. The content and number of the 12 subthemes identified in that research were very similar to the nine scales finally identified at the final phase. Moreover, all these scales had strong item-scale correlations, excellent item scaling success and good internal consistency (Cronbach alpha coefficients of >.7). Therefore, it seems that our qualitative approach was very efficient.

We applied multiple analysis methods (e.g., multitrait scaling analysis) in our quantitative approach. Messer-Misak & Egger (2016) also used a qualitative and quantitative approach to propose a questionnaire regarding the requirements that therapists need to effectively deploy software solutions in the therapeutic process. Even so, the quantitative analysis was only limited to assessing the internal consistency (by Cronbach alpha coefficient) for two scales, without ruling out for example, that items could be intercorrelated. Therefore, our approach appears to be more promising and comprehensive than previous approaches used for questionnaires on EP. We consider our classification by areas and scales to be an advantage, since this facilitates the relationship between attributes of the same theme and makes it easier to understand and apply to future software.

Most of the attributes were expected by the physiotherapists. At least 75% of the physiotherapists expected to find in the PMS all the items on four of the scales in the administrative activities area and “Data entry and issuance of reports” in the clinical care area. This would suggest that the absence of these attributes in PMS could cause great disconfirmation of physiotherapists’ expectations, and likely poor overall perceived quality of the software and dissatisfaction among physiotherapists. Therefore, PMS developers should consolidate such attributes and reinforce their functionalities.

Two scales concentrated most of the attributes with the least expectations. Thus, 18% of the physiotherapists did not expect to find any item of “monitoring quality of care” and 35.5% of “DHI”. In relation to this last scale, previous studies have identified high levels of inexperience with telerehabilitation tools (between 83%–95% of physiotherapists do not use them) (Rausch et al., 2021; Reynolds, Awan & Gallagher, 2021). This could be a possible explanation for this low level of EP. Consequently, since these attributes are not expected in PMS, their presence could generate new needs, becoming a new line of growth for PMS, with special interest in the DHI, which facilitates the improvement of patients in various processes (Rausch et al., 2021).

Our study included physiotherapists with and without experience with PMS, and it showed that both groups had a similar pattern of expectations with the items of the questionnaire. Furthermore, scales had similar reliability in the two groups. That means that the questionnaire could be independently applied to each group and all recommendations cited in previous paragraphs can be applicable both for physiotherapists with and without experience with PMS. This fact is contrary to what was done in previous studies (Messer-Misak & Egger, 2016), where a main limitation was that junior physiotherapists with little work experience participated in the survey and did not analyze differences compared to experienced physiotherapists.

Our findings have important applications. First, to the best of our knowledge, our study is the first to examine EPs towards PMS from a global approach, considering multiple requirements from different aspects (from editable body charts, communication with patients, to payments and stock control). This is essential to be able to develop software that can satisfy EP, so our information would be valuable for the different developers of specific physiotherapy software. Furthermore, these attributes identified for the first time, would allow analyzing the degree of compliance with them in certain software. Thus, future studies should analyze different software present in the market and the presence or not of the attributes identified in our study, preparing guides with recommendations for physiotherapists. Second, our EPQPMS questionnaire is the first validated questionnaire to measure EP, so it could be used as a model to examine EP from other regions or as an example to consider in order to measure other aspects, such as the perceptions of physiotherapists towards their software or the quality of these (i.e., expectations minus perceptions).

Strength and Limitations

The main strength of our study is based on methodological aspects. The questionnaire, generated to examine the EP, was carried out following several phases: first using focus groups for a first approach, second a cognitive pre-test to ensure a good understanding of the questions, and a third phase for its validation with a sufficient sample size. In addition, for the validation analysis of the questionnaire, multiple tests and statistical procedures were used. Finally, the scales were developed and showed similar reliability in both physiotherapists with and without experience with PMS.

Despite our novel contributions, this study is not without limitations. First, the study was carried out in the Spanish region of the Valencian Community and most of the participants were living or working in this region, so caution must be applied when trying to extrapolate the results. Second, as the recruitment method was mass mailing to physiotherapists, an optimal level of control over the sample was not obtained. Third, while support for the validity of EPQPMS was identified, the validity of an instrument cannot be fully established based on a single administration study.

Conclusions

In order to satisfy physiotherapists on the use of PMS, it is first necessary to know their expectations towards the software. Our study provides a valuable questionnaire of EP towards attributes of PMS for clinical care and administrative activities. It also shows a detailed development process, which can help to develop other EP questionnaires including new and relevant attributes in other sites or times. Finally, the results showed that while most physiotherapists had high expectations in all items of each scale, especially those for administrative activities and regardless of whether or not they had experience in PMS, a relevant proportion of physiotherapists had null expectations on the items related with “Digital Health Interventions”.

Supplemental Information

Appendix S1 Items reformulated by pretest

Click here for additional data file.

Appendix S2 Results of the complementary factor analysis

Click here for additional data file.

Appendix S3 Questionnaire in Spanish

Click here for additional data file.

Supplemental Information 4 Dataset

Click here for additional data file.

Supplemental Information 5 Codebook for the dataset

Click here for additional data file.

The authors thank the volunteers for their participation during this study, Jesús Ramírez-Castillo for his collaboration during the study development, and the ICOFCV for the assistance provided to broadcast the study among their members.

Additional Information and Declarations

Competing Interests

Author Contributions

Data Availability

The authors declare there are no competing interests.

Eduardo Arza-Moncunill conceived and designed the experiments, performed the experiments, prepared figures and/or tables, authored or reviewed drafts of the article, and approved the final draft.

Francesc Medina-Mirapeix conceived and designed the experiments, analyzed the data, prepared figures and/or tables, authored or reviewed drafts of the article, and approved the final draft.

Rodrigo Martín-San Agustín conceived and designed the experiments, analyzed the data, prepared figures and/or tables, authored or reviewed drafts of the article, and approved the final draft.

The following information was supplied regarding data availability:

The data and the codebook are available in the Supplemental Files.

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
