# Peer review of "Development and validity of the expectations of physiotherapists questionnaire on practice management software"

_PeerJ, doi:10.7717/peerj.16246_

## Round 0.1 · original submission · Major Revisions

Dear Authors

We have received the revisions to your manuscript. The reviewers consider that your manuscript should not be published in its current version and suggest major changes that I believe should each be addressed in a new version of the manuscript.

Some of the concerns that the reviewers mention are that there is not a sound and well-justified theoretical basis in the introduction and discussion of the manuscript.

I also agree that the initial phase of instrument construction needs to be developed in much more depth and that phase 1 needs to be further explained by describing the literature findings and synthesis of results. In phase two, the qualitative analysis needs to be further explained and the full qualitative results described. Readers should have a deeper understanding of the phases of instrument construction and the results of each phase. This is a new instrument and requires further explanation of all processes.

There are some recommendations on psychometric aspects that need to be addressed.

I consider it important to add the analysis of the floor effect and to state the number of items that have not been answered.

I encourage the authors to address each of the reviewers' requirements; it can help the manuscript improve very significantly.

Reviewer 1 ·

Basic reporting

This study developed the questioner to survey of EP towards a list of attributes for PMS.
These results were relevant the EP items with PMS in modern technologies.
I thought that this questioner is important to the development PMS of EP. However, there are minor revision in introduction and discussion.

Introduction
In previous studies suggested that their overall adoption in the physiotherapy clinics has been low and slow over time. How does development of PMS questioner contribute to the determine factors?

Discussion
Discussions were described the results and no references. Therefore, discussion make citation of the previous studies. In this study, experiences in the use of digital technologies was not significant. Please describe the reason of differences to previous studies.

Experimental design

nothing

Validity of the findings

nothing

Reviewer 2 ·

Basic reporting

Thank you for giving me the opportunity to read this interesting work.

The English used throughout the manuscript seems clear to me, however I'm not a native speaker.

The literature references are insuficient as well as the background to justify the need of the research

The structure of the artice is correct, as well as the tables.

Authors need to improve the quality of figure 1 and 2.

Experimental design

Is the final aim is to improve physiotherapists satisfaction? This is a vague and too general aim that is not achieved with this reserach. It could be a topic to talk about in the discussion but not in the aims of the study.

It seems difficult to understand the real application and the need for this new questionnarie. Probably it is useful for some private technology company to develop new softwares. There are not any health related topics addressed in the questionnaire. Please improve the introduction regarding the need of a questionnarie like this. And improve the discussion explaining the reasons why your questionnaire might be of interest to the scientific community and also the general applicability of the questionnarie.

Validity of the findings

Please explain in more depth and detail the qualitative analysis of Phase 2 in Methods and Results.

Deeply expand the exploratory factorial analysis. There is a methodological problem in that analysis: at the psychometric level it is known that using a principal compenent analysis is not a reliable option since it magnifies the factorial weights of the items and the explained variance. I suggest you use another type of factor analysis such as maximum likehood or principal axes which are much more demanding procedures.

Being the questionnaire such a specific and targeted instrument, I imagine that the study sample was recruited with specific inclusion criteria such as knowledge in digital competencies, experience in software management, etc... If this was not done in this way, it is a major limitation of the study.

There is a lack of references in the statistical analysis to justify the cut-off values being used.

Current psychometric theories suggest that the final factors or subscales should have at least three items, you include a two-item factor, I recommend eliminating them or at least presenting a new factor analysis with 12 items.

Additional comments

Lines 135-136. please include the reference to justify the 0.60 cut-off.

Line 140-141, the sentence has a reference that is not correct for the infromation explained. The referecne is about another study in which another questionnarie is developed, hpwever this sentence should have a reference based in statistics.

Line 151-153. In the same line as the previous paragraph, the reference (Medina-Mirapeix, 2015) does not justifies the sentence.

Line 154-155. The sentence need reference.

References: First and third refereces of the references section are incomplete. Please include the missing information regarding the references' style of the journal.

---

## Round 0.2 · Minor Revisions

Dear Authors,

I commend your diligence in addressing the feedback from the reviewers. However, there remain subtle methodological concerns raised by Reviewer 2, which I deem imperative to the manuscript's clarity and quality.

In relation to the factor analysis, I recommend providing further clarity, either by introducing an additional table or by amalgamating the two analyses suggested during the prior review into the current presentation.

It is imperative that the authors rectify these concerns to ensure the manuscript meets the publication standards of our journal.

Kind regards

Reviewer 1 ·

Basic reporting

no comment

Experimental design

no comment

Validity of the findings

no comment

Reviewer 2 ·

Basic reporting

The authors have done a good job reviewing the manuscript, however there are still some points to be covered:

It is needed a table/s showing the data that the authros expose in the first paragraph of the results. All the statements exposed in results section need a numerical support, (i.e:"Some items were eliminated due to inconsistency,...").

It is not very understandable the wording of the first paragaph of results. It would be interesting to start introducing the "two areas" of the questionnaire and then continue with the explanation of the results for each of the areas.

Another issue is that the authors say they have done the factor analysis of maximum likehood and principal axes, however the results of those analysis are not shown in the manuscript.

Finally, they also say they have analyzed the floor effect, but I can't find it in the manuscript.

Experimental design

Ok

Validity of the findings

Ok

Additional comments

Congratulations for your work.

---

## Round 0.3 · accepted · Accept

Dear authors,

I am delighted to inform you that your manuscript has been accepted for publication. We are grateful for your submission to our journal and hope you will continue to consider us for future studies.